# Heavy Metals Distribution, Sources, and Ecological Risk Assessment in Huixian Wetland, South China

**Liangliang Huang** [1,2], **Saeed Rad** [1,*], **Li Xu** [1,3], **Liangying Gui** [4], **Xiaohong Song** [2], **Yanhong Li** [1], **Zhiqiang Wu** [3] **and Zhongbing Chen** [5,*]

1. College of Environmental Science and Engineering, Guilin University of Technology, Guilin 541004, China; llhuang@glut.edu.cn (L.H.); Foopakdaman@gmail.com (L.X.); sdrd1357@yahoo.com (Y.L.)
2. Guangxi Key Laboratory of Environmental Pollution Control and Technology, Guilin University of Technology, Guilin 541004, China; bpirmoradi@yahoo.com
3. Coordinated Innovation Center of Water Pollution Control and Water Security in Karst Area, Guilin University of Technology, Guilin 541004, China; wuzhiqiang@glut.edu.cn
4. College of Tourism and Landscape Architecture, Guilin University of Technology, Guilin 541004, China; llhuang1986@163.com
5. Department of Applied Ecology, Faculty of Environmental Sciences, Czech University of Life Sciences Prague, Kamýcká 129, 16521 Prague, Czech Republic
* Correspondence: saeedrad1979@gmail.com (S.R.); zhongbing.chen@gmail.com (Z.C.); Tel.: +86-773-253-6372 (S.R.); +42-022-438-2994 (Z.C.)

**Abstract:** This research has focused on the source identification, concentration, and ecological risk assessment of eight heavy metals in the largest karst wetland (Huixian) of south China. Numerous samples from superficial soil and sediment within ten representative landuse types were collected and examined, and the results were analyzed using multiple methods. Single pollution index ($Pi$) results were underpinned by the Geoaccumulation index ($I_{geo}$) method, in which Cd was observed as the priority pollutant with the highest contamination degree in this area. As for the most polluted landuse type, via applying Nemerow's synthetical contamination index ($P_N$) and Potential ecological risk index ($RI$), the river and rape field posed the highest ecological risks, while moderate for the rest. To quantify the drivers of the contaminants, a principal component analysis (PCA) was carried out and weathering of the watershed's parent carbonate rocks was found to be the main possible origin, followed by anthropogenic sources induced by agricultural fertilizer. Considering the impacts of these potentially toxic elements on public health, the results of this study are essential to take preventive actions for environmental protection and sustainable development in the region.

**Keywords:** heavy metals; karst wetland; soil contamination; risk assessment

## 1. Introduction

Both the quantity and quality of our finite freshwater resources [1] for the growing earth population [2] are being menaced by manmade changes in the environment [3]. Global warming and climate change, driven by the greenhouse effect, and uncontrolled production of contaminants are threatening sustainable development. On the other hand, massive landuse/land cover changes since the industrial revolution have caused quicker peak runoffs that ease pollutant transportation. Among different types of contaminants, heavy metals, even at low concentrations, have increasingly caused health concerns due to their hazardous bioaccumulation ability through the food chains [4,5]. Most of these non-degradable toxic elements, such as Arsenic (As), Cadmium (Cd), Chromium (Cr), Copper (Cu), Mercury (Hg), Nickel (Ni), Lead (Pb), and Zinc (Zn), are listed as priority pollutants to control by the EPA [6,7].

As a stormwater control strategy, wetlands are among the highly productive ecosystems that supply habitats and groundwater aquifers (after purification), in addition to peak flood control. However, they are heavily impacted by various pollutants carried by runoff due to the intensive human activities in the landscape [8,9]. Since understanding the nature of every phenomenon is a prerequisite for a sensible attempt to predict its future requirements, the important variables and influencing factors must be measured, analyzed, and monitored. Wetlands, in general, can be polluted by various types of harmful substances, including heavy metals. These transferred elements are in soluble form in water or accumulated in soil/sediments through several pathways, such as atmospheric deposition, sewage, stormwater as well as leachate, which carry contaminants originated from various residential, industrialized, or cultivated areas [9,10]. Hence, sediments as a sensitive reference to monitor pollutants in the environment can provide extensive information on changes in aquatic and watershed ecology [11–14].

Our study area (Guangxi province) is famous for the plentiful mineral resources and is among the top-ten regions of China in nonferrous metal production. It produces 64 kinds of metals in which, for 12 kinds, it is ranked as the number one [15]. Therefore, naturally, in this karst Devonian limestone bedrocks, sizable amounts of heavy metals were observed, especially in the soil, owing to its backgrounds, geologically [16,17]. Some other researches related the increased heavy metals in soil and sediment to the mining and smelting activities in this region [17–19]. Moreover, the contents of heavy metals in farmland posing a potential risk to human health have been raised here [20–24]. For better protection, Huixian has been listed as a National Wetland Park since 2012. However, although the wetland is impacted by heavy metals [25,26], the detailed distribution characteristics in the soil and sediments are not clearly studied yet, and so their potential ecological risk is poorly explored. Therefore, the goals of the current work are: (1) investigating the distribution and concentration of eight types of heavy metals in the wetland; (2) to identify the possible sources of these contaminants; (3) to find out the most impacted areas among 10 different types of landuse; (4) and lastly, the heavy metals potential risks assessment, ecologically in this region. The outcomes of this study will help the decision-makers for environmental monitoring and efficient local pollutant management strategies.

## 2. Materials and Methods

### 2.1. Study Area

Huixian wetland, located in the northeast of Guangxi, with an area of 587 hm$^2$, is the greatest wetland of karst areas in China. It forms a complex ecosystem consisting of lakes, marsh, rivers, ponds, and artificial canals [27,28], as shown in the map presented in Figure 1. The mean annual temperature is 16.5–20.5 °C, and the average rainfall 1890 mm, yearly. Since the 1950s until 2012, when it was listed as a national wetland park, the wetland has been damaged. These damages were especially landuse changes and conversion of natural lands to aqua/agricultural lands continually, because of the increased manmade changes, the poor management, as well as lack of efficient protections. These have gradually caused the wetland water areas and its connected ponds shrinkage, and therefore, the ecosystem of the wetland has been critically impacted. Currently, the entire area covered by water is just around one square kilometer [28]. The large grassy areas around the wetland have been recently opened up to about 130 hm$^2$ of farmland, and the marshes have been excavated to 280 hm$^2$ to convert to fish farms [26].

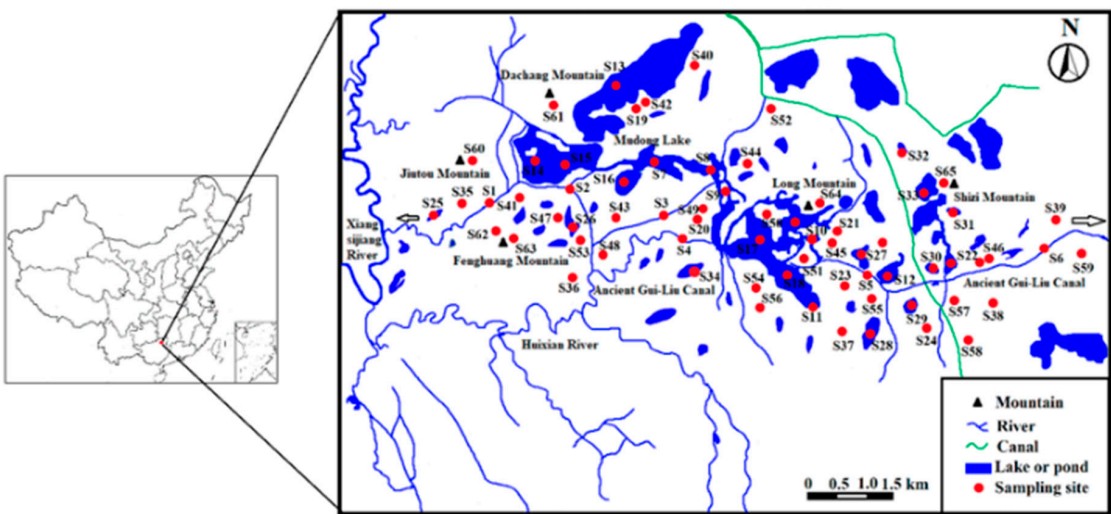

**Figure 1.** Sampling sites in the Huixian wetland.

*2.2. Sample Collection and Testing Procedures*

A total of 65 surface topsoil and sediment samples (31 and 34, respectively) were collected from 10 representative landuse types of the Huixian wetland, as shown in Figure 1. Using a Peterson borrow device, samples from the arable layers of the soil (top to 10 cm depth) were collected at the maize field (S35–S40), paddy field (S41–S46), rape field (S47–S52), vegetable field (S53–S59), and woodland (S60–S65). Also, the sediment samples of the top layer (0–10 cm) were obtained from the river (S1–S6), marsh (S7–S12), lake (S13–S18), agricultural ditch (S19–S24), and fish pond (S25–S34). In each sampling site, three replicated sub-samples were taken and mixed carefully via hand to acquire composite samples. All the mixtures were then placed into a polyethylene bag to avoid any possible external impacts (to keep them separated, proper labeling process, rainfall effects, etc.) and transferred to the lab.

In the lab, each sample was air-dried at room-temperature before passing through 20 meshes, for the removal of bigger fragments and rubble. The soil samples were grounded after this step via mortar and pestle until all particles passed 100 meshes [11]. For Cu, Cr, Cd, Ni, Zn, and Pb, about 0.5 g dry sediment/soil of each sample was digested in a microwave oven with a mixture of acid $H_2O_2$-$HNO_3$-HF-HCl (120 °C/3 min-150 °C/5 min-170 °C /25 min). For As and Hg, about 0.1 g dried amount of each sample was digested using mixed acid of $HNO_3$ and HCl at 100 °C for 2 h and then diluted to 25 mL with distilled water before analyses. Cd was tested by the graphite furnace method via atomic absorption spectrometry or AAS (PerkinElmer PinAAcle 900T, Boston, MA, USA) with a detection limit of 1.5 ug/L. However, coupled plasma method using optical emission spectrophotometer or ICP-OES (PerkinElmer optima 7000DV, USA) applied to measure the real concentration of Cu, Cr, Ni, Pb, and Zn with detection limits of 0.4, 0.2, 0.5, 1, 0.2 ug/L, respectively. Hg and As measured via atomic fluorescence spectrometer or AFS (Beijing Jitian SA-20A, Beijing, China) with detection limits of 0.01 and 0.001 ug/L [29–31]. For quality control/assurance purposes, appropriate handlings were followed throughout the entire process (from sampling to analysis). The QA/QC was in order to avoid probable cross-contaminant and also to control the ambient condition (such as temperature and humidity) to gain accurate results. Duplicate, blank solution, standard material (according to certified reference material GBW07404 for all the contaminants), and additional recovery, all were conducted, attentively. The relative standard deviation (RSD) ranged from 1.12%–6.03% (n = 3), and recoveries were in the range of 92% to 108% for different contaminants, accordingly.

*2.3. Data Analysis and Source Identification*

A one-way ANOVA test was applied to compute whether if there were significant differences in results obtained from different landuse types. Using SPSS 18.0, a Pearson correlation analysis was performed to specify the possible correlation among variable series that were measured in the current work. Inter-relationships between different heavy metals could offer witting on possible resources and their trajectories [32]. This relationship can be recognized from a coefficient (*r*) value, which shows the strength level of association between two variables. Where *r* is greater than 0.7, between 0.4 and 0.7, or if it is smaller than 0.4, the quantified variables would have high, moderate, or weak correlations, respectively [33]. The mean values also were calculated as the representative of the entire wetland in order to compare with the background values, as well as standard levels.

For source identification, Principal Component Analysis or PCA, with Varimax normalized rotation was employed, separately. This method was used to determine the possible concealed relationship on the obtained results, which can reduce the original multi-dimensional spaces of PCs (the principal component). The PCA calculates the eigenvector values in order to carry out similar contaminants' origins. For this analysis, the relevance is identified as the components with eigenvalues of bigger than 1.0 [34], a component that has factor loading > 0.75, from 0.75–0.5, and from 0.5 to 0.3 is taken as strongly, moderately, and weakly relevant, respectively [35]. Besides applying the Boxplot, which is a proper method to describe the distributions of the data and comparison of various landuse types' contaminants, these were also plotted on the study area's map using CAD (computer-aided design).

*2.4. Risk Assessment*

To measure the contaminations risk based on the results and to identify the potentially contaminated regions, three major risk assessment methods were employed in this study, as shown below:

1.  *Geoaccumulation index* ($I_{geo}$), which is determined via the bellow formula [36].

$$I_{geo} = \log_2[C_i/1.5B_i] \tag{1}$$

where $C_i$ is the heavy metal real concentration in the studied site; and $B_i$ would be the reference sample background value (Table 1) [37]. Generally, the $I_{geo}$ consists of 7 grades in the range of $5 < I_{geo} \leq 0$ in which minimum values indicate the soil has not been contaminated, while maximum values show it has been extremely contaminated [38–40]. In fact, $I_{geo} \leq 0$ means that the soil is not contaminated; $0 < I_{geo} \leq 1$ indicates uncontaminated up to moderately contaminated degrees; $1 < I_{geo} \leq 2$ presents a moderately contaminated degree; $2 < I_{geo} \leq 3$ means moderately up to strongly contaminated degrees; $3 < I_{geo} \leq 4$ indicates a strongly contaminated degree; $4 < I_{geo} \leq 5$ presents strongly up to extremely contaminated degrees, and lastly $I_{geo} > 5$ shows that the soil has been extremely contaminated.

2.  *Nemerow's synthetical contamination index* ($P_N$) is calculated via this formula as follows:

$$P_N = \sqrt{\frac{AvgP_i^2 + MaxP_i^2}{2}} \tag{2}$$

where $P_N$ was the Nemerow's synthetical contamination index, which indicates the contaminant's gradation; $P_i$ is the single pollution index, which is proportionally related to the *directly quantified concentration* $C_i$ of the contaminant *i* in each site, with the pollutant concentration standard value of $S_i$. Hence $P_i = C_i/S_i$; and the $MaxP_i$ and $AvgP_i$ would be the maximum and the average values of all the indices $P_i$, respectively. The $P_N$ consists of 5 grades from $P_N \leq 0.7$ up to $P_N > 3$. In details, the $P_N \leq 0.7$ is the safety domain; $0.7 < P_N \leq 1$ is precaution domain; $1 < P_N \leq 2$ is slightly polluted domain; $2 < P_N \leq 3$ is moderately polluted domain; and finally the $P_N > 3$ is seriously polluted domain [41–43].

3. *The potential ecological risk index* (*RI*) as the summation of the all measured heavy metal's potential ecological risk factors ($E^i_r$), calculated as below:

$$RI = \sum_{i=1}^{m} E^i_r \tag{3}$$

in which four categories of the potential ecological risk index were identified here as: *RI* less than 150 would have low risk; between 150 to 300 for moderate risk; between 300 to 600 as high and lastly; greater than 600 would have very high risk. Also for the potential ecological risk factor ($E^i_r$) five categories are identified, including $E^i_r$ less than 40 as low; between 40 to 80 to be considered as moderate; between 80 to 160 as considerable; from 160 to 320 as high; and for $E^i_r$ greater than 320 as very high potential ecological risks. This factor ($E^i_r$) was computed using the following equation:

$$E^i_r = T^i_r \times [C^i / B^i_n] \tag{4}$$

where $C^i$ is the metal *i* concentration in the sediment sample, $B^i_n$ is the metal *i* background value, and $T^i_r$ is the metal *i* toxicity index in which in this formula are: 1, 2, 5, 5, 5, 10, 30, and 40 for Zn, Cr, Cu, Ni, Pb, As, Cd, and Hg, respectively [44,45].

**Table 1.** Heavy metal concentrations (mg/kg in dry weight) in samples taken from different landuse types of the wetland.

| Land-Use Types | # of Samples | As | Cd | Cr | Cu | Hg | Ni | Pb | Zn |
|---|---|---|---|---|---|---|---|---|---|
| River | 6 | 20.81 ± 1.69 | **0.96 ± 0.43** | 105.73 ± 3.89 | 34.33 ± 5.13 | **0.52 ± 0.02** | **49.50 ± 1.54** | 55.80 ± 1.05 | 122.07 ± 17.90 |
| Marsh | 6 | 18.61 ± 3.24 | **0.90 ± 0.23** | 100.20 ± 5.84 | 35.53 ± 5.64 | 0.27 ± 0.09 | 45.43 ± 3.64 | 67.63 ± 3.53 | 112.37 ± 14.37 |
| Lake | 6 | 14.00 ± 1.86 | **1.01 ± 0.55** | 111.67 ± 33.75 | 25.07 ± 10.76 | 0.18 ± 0.10 | 36.47 ± 7.68 | 51.37 ± 10.39 | 107.70 ± 18.30 |
| Agricultural ditch | 6 | 17.70 ± 4.36 | **0.49 ± 0.15** | 95.05 ± 33.23 | 36.78 ± 28.87 | 0.15 ± 0.08 | 28.33 ± 12.28 | 53.08 ± 16.14 | 82.90 ± 28.19 |
| Fish pond | 10 | 17.52 ± 4.23 | **0.58 ± 0.22** | 122.18 ± 25.85 | 39.53 ± 22.81 | 0.15 ± 0.13 | 40.85 ± 9.08 | 38.14 ± 20.50 | 138.41 ± 67.34 |
| Maize field | 6 | 18.42 ± 2.24 | **0.72 ± 0.14** | 126.77 ± 18.27 | 31.80 ± 2.63 | 0.29 ± 0.29 | 44.27 ± 4.35 | 47.57 ± 16.14 | 115.43 ± 6.83 |
| Paddy field | 6 | 13.93 ± 2.13 | **0.69 ± 0.27** | 121.98 ± 20.07 | 31.45 ± 3.81 | 0.14 ± 0.14 | 41.93 ± 3.39 | 50.55 ± 20.70 | 111.47 ± 10.60 |
| Rape field | 6 | 20.33 ± 5.41 | **0.95 ± 0.10** | 117.77 ± 14.26 | 34.13 ± 0.68 | **0.49 ± 0.37** | 37.33 ± 3.84 | 37.00 ± 6.37 | 109.35 ± 0.45 |
| Vegetable field | 7 | **29.58 ± 7.97** | **0.31 ± 0.23** | 125.71 ± 19.47 | 51.97 ± 18.02 | 0.16 ± 0.10 | **59.23 ± 14.90** | 43.30 ± 23.77 | 138.91 ± 26.58 |
| Woodland | 6 | 16.84 ± 5.36 | **0.30 ± 0.08** | 124.53 ± 24.69 | 24.53 ± 6.53 | 0.08 ± 0.06 | 44.63 ± 4.51 | 70.47 ± 12.51 | 128.60 ± 30.05 |
| Mean of all | 65 | 18.86 ± 6.14 | **0.62 ± 0.31** | 118.18 ± 23.55 | 37.25 ± 18.23 | 0.20 ± 0.18 | 43.04 ± 11.23 | 46.59 ± 19.71 | 124.39 ± 46.17 |
| SEPA limitation * | - | 25 | 0.3 | 300 | 100 | 0.5 | 50 | 300 | 250 |
| Background value ** | - | 10.82 | 0.19 | 70.18 | 23.78 | 0.13 | 23.37 | 29.95 | 72.61 |

Notes: Bold values indicate higher than limit; * State Environment Protection Administration of China [46]; ** Data collected from Zheng (1993) [37].

## 3. Results and Discussions

### 3.1. Heavy Metals Concentrations

The heavy metal's concentration obtained in samples is demonstrated in Table 1 and evaluated in accordance with the National Quality Standard for Soil in China (GB15618-1995). Results revealed that, except for Cd, which was higher than the standard level in all the landuse types, other elements in the majority of landscapes were within the acceptable limits of Class II, for environmental protection. ANOVA analysis shows how significantly the concentration of each pollutant is varied in different landuse. As can be seen, besides Cd, another three heavy metals include Hg (in the river and rape field), Ni (in the river and vegetable field), and As (in the vegetable field, where river and rape field were the second highest), had remarkably greater values than the rest of the collected samples. Furthermore, there were no considerable differences observed for the mean concentration for Zn, Cr, Cu, and Pb.

These results show two major facts. First that the most number of elements with values higher than the standard levels were found in the sediment (river followed by the lake and marsh) compared to the soil. This is possibly due to the long-term settlements of sediment-bound contaminants, with less disturbance compared to the soil (such as tillage, sunlight illumination, plant uptake). This will make these sediment banks like a contaminant bank during the time, which can supply heavy metal pollutants to the farms and the environment with every resuspension or overflow resulted from runoff and flood. Second, that the Cd was the dominated heavy metal among the eight measured contaminants, which must be focused on in future researches on this wetland.

Comparatively, average values that resulted from As, Cr, Cu, and Hg in Huixian were between 30% to 50% higher than the mean concentrations of these contaminants in sediment in nine other wetlands in China, including those in Guangxi province (Table 2). However, Ni, Pb, Zn were at the same level with the rest of the areas, in which Cd was around 70% lower than in other regions, on average (except for the extreme values) [9–14,21–24,29,38,47]. Although Cd was the highest element in all landuse types of the Huixian wetland, nevertheless, it was among the lowest averages as compared to other studies in different parts of the country. This is most probably owing to the strong background value of Cd (0.19 mg/kg) versus the standard limits (0.30 mg/kg). It means that mineralogically, this region has a large amount of Cd naturally, and hence, the minimum manmade changes (such as applying fertilizers or industrial sources like still, plastic, and batteries [48]) can cause exceeding the threshold.

**Table 2.** Comparison between heavy metal concentrations (mg/kg) found in sediment in this and the previously published studies of Chinese wetlands (mean values).

| Wetland | Province | As | Cd | Cr | Cu | Hg | Ni | Pb | Zn | Reference |
|---|---|---|---|---|---|---|---|---|---|---|
| Caohai Natural Wetland | Guizhou | | **26** | | 27 | | | **99** | **540** | [29] |
| Dongting Lake Wetland | Hunan | **25.67** | 4.39 | 91.33 | 36.27 | 0.19 | **46.36** | 54.82 | | [12] |
| Han River Wetland | Shanxi | | 0.56 | 84.1 | 38.7 | | 38.6 | 23.5 | 94 | [10] |
| Poyang Lake Wetland | Jiangxi | 6.69 | 0.42 | 105.77 | 12.25 | | 30.47 | 27.81 | 79.45 | [9] |
| Wulihu Lake Wetland | Jiangsu | | 4.08 | **261.2** | 14.28 | | | 52.4 | 268.1 | [47] |
| Yellow River Wetland | Henan | | 0.11 | 53.6 | 39.3 | | | 41.1 | 72.4 | [38] |
| Yilong Lake Wetland | Yunnan | 15.46 | 0.76 | 86.73 | 31.4 | | 35.99 | 53.19 | 86.82 | [11] |
| Yongnianwa Wetland | Hebei | | | 71.69 | **43.3** | | 42.54 | 44.14 | 118.95 | [13] |
| Zhalong Wetland | Heilongjiang | 10.26 | 0.155 | 46.47 | 18.17 | 0.065 | | 21.38 | 52.09 | [14] |
| Mean of Chinese Wetlands | - | 14.52 | 4.56 | 100.11 | 28.96 | 0.13 | 38.79 | 46.37 | 163.98 | |
| Huixian Wetland * | Guangxi | 17.62 | 0.67 | 114.67 | 37.12 | **0.20** | 40.16 | 45.22 | 125.43 | Current study |

* Average concentration in sediment samples. Bold values indicate higher than limit

### 3.2. Principal Component Analysis and Correlation Matrix

The Pearson correlation matrix showed that Ni and As are highly correlated with an *r*-value of 0.751 (Table 3). Furthermore, moderate correlations were found between Cr with As, Ni, and Cd, as well as between Zn and Cu, while the rest, such as Hg with Cd, were weekly correlated.

**Table 3.** Correlation matrix for *r* values of heavy metals in soil and sediment from the Huixian wetland.

| Heavy Metals | As | Cd | Cr | Cu | Hg | Ni | Pb | Zn |
|---|---|---|---|---|---|---|---|---|
| As | 1.000 | −0.306 * | **0.432 **** | 0.252 * | 0.224 | **0.751 **** | 0.053 | 0.239 |
| Cd | | 1.000 | **−0.485 **** | 0.008 | 0.384 ** | −0.291 * | 0.132 | 0.003 |
| Cr | | | 1.000 | 0.045 | −0.110 | **0.595 **** | −0.279 * | 0.050 |
| Cu | | | | 1.000 | 0.003 | 0.289 * | −0.273 * | **0.446 **** |
| Hg | | | | | 1.000 | 0.236 | 0.023 | −0.054 |
| Ni | | | | | | 1.000 | −0.077 | 0.203 |
| Pb | | | | | | | 1.000 | −0.114 |
| Zn | | | | | | | | 1.000 |

* and ** Correlation is significant at the 0.05 and 0.01 level (2-tailed). Bold values are moderately/highly correlated.

Moreover, the PCA analysis yielded three significant components with eigenvalues higher than 1.00, accounting for a total of 67.64% of the data variation (Table 4). The first principal component (PC1), which contained 26.68% of the calculated variance, showed a strongly positive load for As and Ni, but moderately for Cr. Chromium and Nickel are known to be mutually associated with several sorts of rocks and, hence, into any soil driven of these strata [49]. The same was reconfirmed in our research, obtaining a correlation coefficient of r = 0.595 between them. Besides the anthropogenic sources of Cr and Ni from fertilizer, both the limestones or manures have lower concentrations as compared to those in the soil samples [33]. In addition, the As was positively related to Cr (r = 0.432) and Ni (r = 0.751), which indicates that As might be driven from parent rock materials.

**Table 4.** Principal Component Analysis for heavy metals in sediments and soils from the Huixian wetland.

| Heavy Metal | PC1 | PC2 | PC3 |
|---|---|---|---|
| As | 0.89 | - | - |
| Cd | - | - | 0.64 |
| Cr | 0.65 | - | - |
| Cu | - | 0.75 | - |
| Hg | - | - | 0.75 |
| Ni | 0.92 | - | - |
| Pb | - | - | 0.56 |
| Zn | - | 0.63 | - |
| Proportion of Variance (%) | 26.68 | 23.03 | 17.93 |
| Cumulative Proportion of Variance (%) | 26.68 | 49.71 | 67.64 |

Note: factor loadings < than 0.5 removed, extraction method: PCA, rotation method: Varimax and Kaiser normalization.

The second principal component, which explained 23.03% of the measured variance, showed a strongly positive load of Cu, but moderately positive loading of Zn. These two elements in the natural soils are affirmed to show close geo-chemical dependence as the iron family [49], which is presented again in the current results with a correlation coefficient of r = 0.446. The third principal component, however, for 17.69% of the obtained variance, found of a strongly positive load for Hg, but moderately positive load for Cd and Pb. Taking into consideration the Cd (as an utmost toxic element which has great ecological risks) correlation to Hg, our study suggests that these pollutants are probably driven by long-term use of phosphate fertilizers [50] beside high natural backgrounds. Commercial phosphate fertilizer containing small amounts of different elements (such as Zn, Hg, Cd,

and Pb) originated from its raw materials, are exhibited as important sources of such compared to other inorganic fertilizers [51,52].

### 3.3. Risk Assessment

Based on the geoaccumulation index method, the results of $I_{geo}$ values for eight elements ranged from −1.28 to 1.83. The average degree of pollution for each heavy metal decreased as per the following order: Cd > Ni > As > Pb > Cr > Hg > Zn > Cu. Most of the quantified heavy metals in every landuse had values of less than 1.0 (uncontaminated to a moderately contaminated degree), except for Cd in seven sites and Hg in two sites in which the amounts were greater than 1.0, which means moderately contaminated (Figure 2). The average contamination degree of Hg in the entire wetland was not serious as per the $I_{geo}$ results due to its negative values in most of the landuse types.

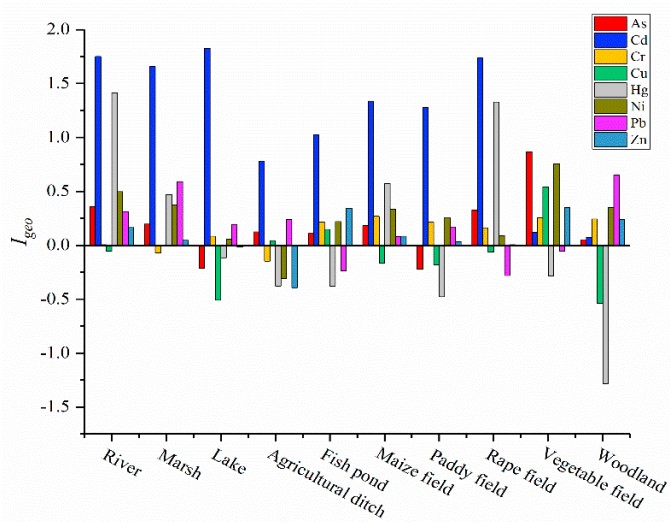

**Figure 2.** Geoaccumulation index ($I_{geo}$) of eight heavy metals in different landuse types in the wetland.

As per the single pollution index ($P_i$) however, the trend of the measured heavy metals was distributed in decreasing order as Cd > Ni > As > Hg > Zn > Cr > Cu > Pb (Figure 3). The highest obtained results were for Cd in every landuse type, except for woodland.

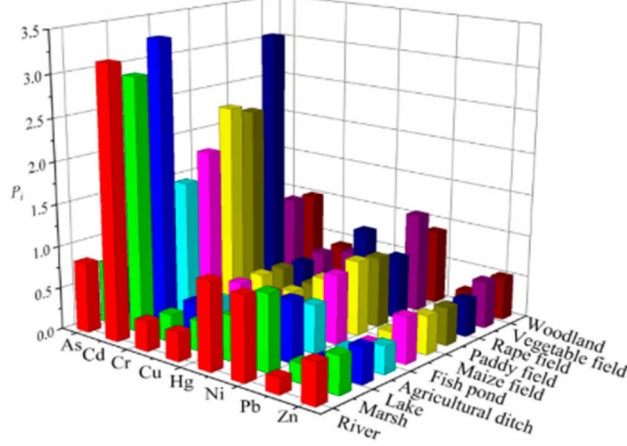

**Figure 3.** Single pollution index ($P_i$) of heavy metals in different landuse types in Huixian wetland, colors representing landuse types.

The Nemerow's synthetical contamination index ($P_N$) result, which is based on the single pollution index ($P_i$), revealed that the river, marsh, lake, and rape field were moderately contaminated with

$P_N$ values of higher than 2.0, but the fish pond, maize field, paddy field, and agricultural ditch were slightly contaminated with $P_N$ values of higher than 1.0 (Figure 4).

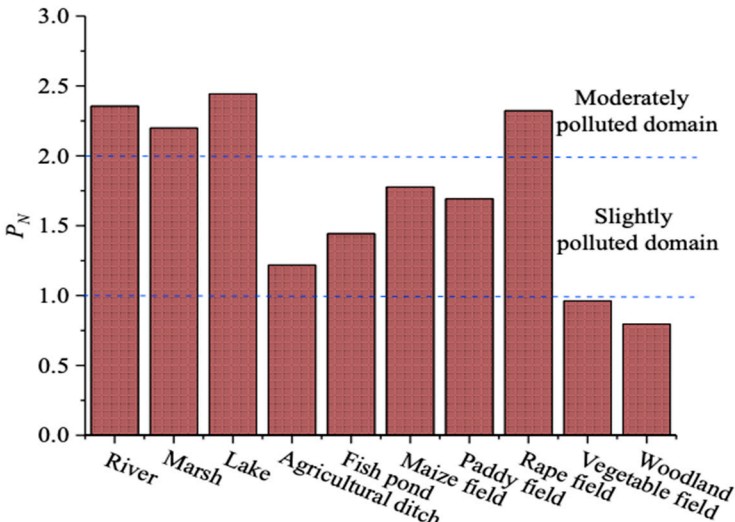

**Figure 4.** Nemerow's synthetical contamination index ($P_N$) in different landuse types of the wetland.

The potential ecological risk index (*RI*) calculated for contaminants in our work is shown in Figure 5. Samples from the river and rape field were found to pose the highest ecological risk ($300 \leq RI < 600$), while in woodland, showed the lowest ecological risk ($RI < 150$), and the rest posed a moderate risk ($150 \leq RI < 300$). According to the *RI* index, Cadmium was found to have a considerable potential ecological risk ($80 \leq E^i_r < 160$) in the river, marsh, lake, fish pond, maize field, paddy field, rape field, while moderate risk ($40 \leq E^i_r < 80$) in the agricultural ditch, vegetable field, and woodland. Also, Hg was shown to pose a high potential risk ($E^i_r$ between 160 to 320) in the river, considerable risk ($E^i_r$ between 80 to 160) in the marsh, maize field, and rape field, but moderate ($40 \leq E^i_r < 80$) in the rest of the landuse types. In addition, all other heavy metals posed a low potential risk ($E^i_r < 40$) in different landscapes.

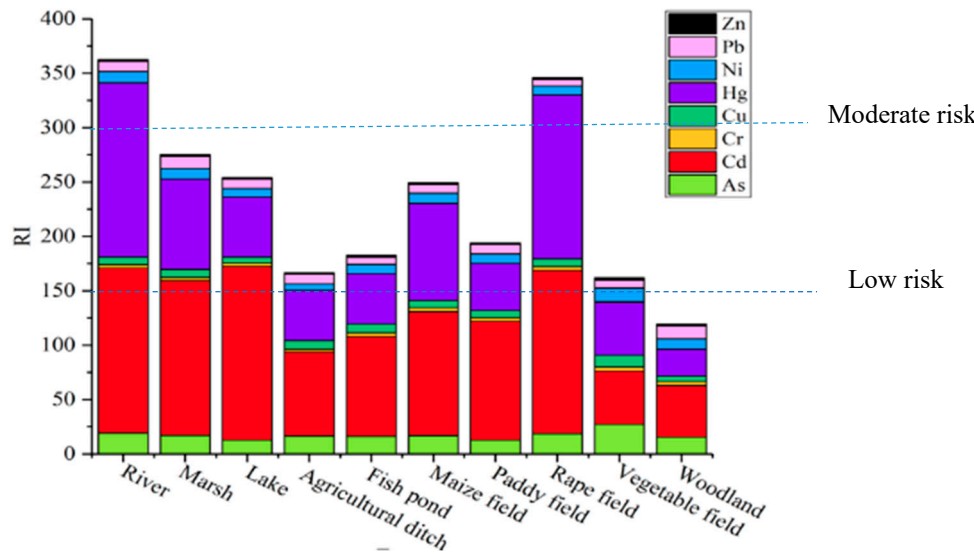

**Figure 5.** Ecological risk indices (*RI*) in different landuse types of the Huixian wetland.

On average, sediment samples taken from water-related landuses (river, lake, etc) were 11% more polluted than soil samples collected from different farms or woodland, and specifically, Cd was found

to be approximately 55% greater in the sediment samples compared to the soil. One of the reasons behind this might be generally due to the plant's heavy metal uptake ability, which has a negative impact on the pollutant accumulation process [53]. The different levels of contaminants in different farms is also possibly due to different retention abilities of various kinds of plants in heavy metal uptake. Although the results of several risk assessment methods in this work were not exactly identical, the $I_{geo}$, $P_i$, $P_n$, and RI indexes all indicated that Cd was the most serious contaminant in the wetland among all the measured elements. According to WHO [48], Cd has toxic effects on humans and mainly accumulates in kidneys with a relatively long biological half-life for lower than 35 years. Low amounts of these contaminants can be found in vegetables and cereals.

Moreover, the Ni and As, on average, were the closest to the standard limit (86% and 75%, respectively). This was followed by Hg in pollutant ranking, while Hg posed high-to-moderate ecological risks. The outcomes of these assessments were also in agreement that the river and rape field were the largely affected landuse types where woodland was the least contaminated area with the lowest ecological risk. This is important because the rape field productions are mainly used to produce oil, taking into consideration the oily food culture in this province. With regards to the sources, in addition to the lithological abundance of heavy metals in this watershed, the landscapes could be influenced by human activities (fertilizer application) since the surrounding areas of the wetland, as a farming area, having several decades of intensive tillage, fertilizer, and pesticide application [54].

As Huixian wetland is one of the main bases for agricultural supply in Guilin city, maintaining and improving the quality of edible products is highly important for human health protection and economic developments, locally. Hence preventive BMPs and treatment methods, such as seasonal remediation (after main contaminants identification), must be prioritized by decision-makers to avoid/minimize environmental damages. We also suggest conducting comprehensive research on the amount of heavy metals in different aquatic and agricultural product's tissues in this area to examine the level and harmfulness of such elements existing for public health concerns. The environmental standards (SEPA), which were made to promote ecology and improve people's health, besides social and economic developments, can limit the health risks to some extent. Scenarios like the Huixian wetland shows that parallel with strengthening the standards, monitoring mechanisms are equally important to guarantee the application and maintaining the permitted limits.

## 4. Conclusions

Analytical tools such as $I_{geo}$, $P_i$, $P_N$, RI, and PCA were used to assess the distribution characteristics and magnitude of toxic elements in the Huixian wetland. Eight different heavy metals in 10 landuse types were sampled to identify their possible sources as well as the most contaminant areas with a high ecological risk. In general, heavy metals in the wetland were found to pose a moderate to high potential ecological risks based on our study, in all landuse types except woodland. The highest ecological risks were specifically found in the river and rape field. Sediment samples presented to be 11% more polluted than soil samples, possibly due to long-time sedimentation, less disturbance compared to the soil as well as plant uptakes. According to the PCA results, the sources of the pollutants were mainly originated from the mineralogical background (limestone bedrocks) followed by anthropogenic activities (fertilizers), mutually. The obtained concentration results for all the elements in Huixian were found to be greater than the background values of this region. However, compared to most of the other places in China, the average values in Huixian were higher only for As, Cr, Cu, Hg, but far lower for Cd. Based on the risk assessment results, for two of the contaminants (Cd and Hg), the concentration levels exceeded the grade II of environmental quality standard classification, which suggests a severe to moderate ecological risk and, therefore, needs to be considered as the priority pollutants for the studied site.

**Author Contributions:** Data curation, L.X.; formal analysis, L.G.; methodology, X.S.; software, Y.L.; supervision Z.W. and Z.C.; writing—original draft, L.H.; writing—review and editing, S.R. and Z.C. All authors have read and agreed to the published version of the manuscript.

**Funding:** This research was funded by the Natural Science Foundation of Guangxi (2018GXNSFAA281022, 2016GXNSFAA380104).

**Acknowledgments:** The authors are grateful to the Guangxi 'Bagui Scholar' Construction Project (2016A10) and Guangxi Science and Technology Planning Project (GuiKe-AD18126018).

**Conflicts of Interest:** The authors declare no conflict of interest.

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
