# Peer review of "Heavy Metals Distribution, Sources, and Ecological Risk Assessment in Huixian Wetland, South China"

_water, doi:10.3390/w12020431_

Round 1

Reviewer 1 Report

Dear Author,

I appeciated the topic of the manuscript but I think that some changes are necessaries.

Here, you can find my comments:

Linguistic revision is needed, especially to fix a fair amount of problem with punctuation.

Lines 42-43: Please, put the acronyms into brackets and write the extended name of the elements

Line 48: the sentence lacks the subject (it is written directly “is…”)

Line 73: The coordinates indicates a specific point, not an area.

Lines 75-76: Which kind of changes occurred in the area? Please, describe the damages

Line 94: every sample: every is referred to person/animal and not to thing

Lines 94-108: Please carefully check the language used in this paragraph, as a lot of syntax errors are present.

Lines 133 and 141: Please specify the grade limits.

Line 160: Please, change “apart from” with “except for”

Line 166 and elsewhere in the text: it is not clear whether median or mean was calculated, and why. This should be stated in the Material and Methods Section.

Line 176: Please, write “for” after except

Line 192 and elsewhere in the text: Please correct Nickel spelling. Authors alternate the use of element symbols and their extended name. Please use the same notation through the manuscript.

Results and Discussion Section: While the Authors described extensively and in details the results, discussion has been somehow neglected. The quality of the paper would be greatly improved by the discussion of the meaning of the obtained results, the possible consequences and the potential strategies to avoid contamination, also by referring to literature, considering that the Authors, in the Introduction Section, speculated about the relation between anthropic activities and heavy metal contamination. This instance should be widened in the Discussion based on the observed contamination level and the evaluation of risk.

Fig. 1: considered the potential interest of mining activities, probably the indication of mines could be important.

Tables 1 and 2: Is average used as a synonym of mean?

Table 2: Please specify if values from references are mean or specific values.

Author Response

Comments and Suggestions for Authors

Linguistic revision is needed, especially to fix a fair amount of problem with punctuation.                                                                                              Noted and done accordingly.

Lines 42-43: Please, put the acronyms into brackets and write the extended name of the elements.                                                                                        Done, new line number/s:  42, 43

Line 48: the sentence lacks the subject (it is written directly “is…”)                                                             Amended, new line number/s:  50-52

Line 73: The coordinates indicate a specific point, not an area.                                                                  Amended (removed) , new line number/s:  77

Lines 75-76: Which kind of changes occurred in the area? Please, describe the damages.                                                                                                Added, it is especially landuse changes and conversion of natural lands to aqua/agricultural lands, new line number/s:  80, 81

Line 94: every sample: every is referred to person/animal and not to thing.                                             Amended, new line number/s:  109

Lines 94-108: Please carefully check the language used in this paragraph, as a lot of syntax errors are present.                                                                       Amended, new line number/s:  113-131

Lines 133 and 141: Please specify the grade limits.                                                                                                Done, new line number/s:  277-281, 287-289

Line 160: Please, change “apart from” with “except for”                                                                                      Done, new line number/s:  319

Line 166 and elsewhere in the text: it is not clear whether median or mean was calculated (is mean and corrected), and why (it was calculated to represent the entire wetland contamination degree compared to the standard level and the background values). This should be stated in the Material and Methods Section. Done, new line number/s:  140, 325,339 as well as table 1 and 2

Line 176: Please, write “for” after except                                                                                                                 Done, new line number/s:  461

Line 192 and elsewhere in the text: Please correct Nickel spelling.  Done Authors alternate the use of element symbols and their extended name. Please use the same notation through the manuscript.                                                              Done except for limited cases due to repetition avoidance, new line number/s:  476, 479,..

Results and Discussion Section: While the Authors described extensively and in details the results, discussion has been somehow neglected. The quality of the paper would be greatly improved by the discussion of the meaning of the obtained results, the possible consequences and the potential strategies to avoid contamination, also by referring to literature, considering that the Authors, in the Introduction Section, speculated about the relation between anthropic activities and heavy metal contamination. This instance should be widened in the Discussion based on the observed contamination level and the evaluation of risk.              Duly noted and such discussions are added in 15 new lines: 326-333, 446, 558-560, 565-566, 574, 577-581 as well as in conclusion section.

Fig. 1: considered the potential interest of mining activities, probably the indication of mines could be important.                                                                                                                                                                      There is no mining activities around this wetland but in the watershed, and hence considering the map scale it cannot be added in Fig.1.

Tables 1 and 2: Is average used as a synonym of mean?                                                                                           Yes and the word mean is now used for both tables.   

Table 2: Please specify if values from references are mean or specific values.                                                       All values from the references are mean values, and now specified in the table.   

Reviewer 2 Report

Review of “Heavy Metals Distribution, Sources, and Ecological Risk Assessment in Huixian Wetland, South China”

General comments: The paper covers an interesting and relevant topic; the environmental risk assessment of wetlands. The procedure used seems adequate, with a large number of samples collected and analysed, and the use of indicators to perform the risk assessment. However, the quality of English language makes it difficult to fully comprehend the article. A thorough revision by a native English speaker is required. The discussion section should also be enhanced, see specific comments for questions. Some conclusions require scientific evidence to prove they are more than speculations. For these reasons, I recommend to accept the paper with major revision.

Specific comments:

Lines 46-54: Include more references on wetlands and their use to monitor pollution.

Line 74 (and elsewhere): Do not refer to the position of the figure relative to the text (below), as the figure may be placed elsewhere in the final version.

Lines 78-79: Does the National Wetland Park status preclude other land uses? How can such a protected area be as damaged?

Line 87: Why is the sample depth different for soil and sediments?

Line 90: What are external impacts?

Lines 101-102: Specify which type of digestion was used for which elemental analysis.

Line 105: What do you mean by “control the ambient conditions”?

Line 106: The reported relative standard deviation corresponds to which analysis, or which contaminant?

Line 133: Specify the limits for each category.

Line 139: Is the “pollutant concentration standard” the background concentration?

Line 141: Specify the limits for the three other categories.

Lines 153-155: Use the same symbols as in the other equations, it would be easier for the reader to follow.

Line 162: The ANOVA results are not shown.

Line 166: The median concentrations are not shown.

Line 177: “highest element in the Huixian wetland”: no, other elements are much higher!

Line 180: “magnificient”: wrong word

Lines 192 and 195: Nickel, not nickol

Lines 195-196: “fewer concentrations”, wrong words

Line 205: Too many capital letters

Figure 5: Add category thresholds on graph (similar to Figure 4).

Lines 254-261: Discuss the relationship (or lack of) between risk and SEPA limits. Do the limits help protect sufficiently against environmental risks?

Lines 262-268: How does the Park status help in management and remediation prioritization?

Line 272: “identify their possible sources”: the sources were not discussed much, only speculations.

Line 276: “due to plant uptake”: not demonstrated

Line 277: “anthropogenic activities”: speculated

Author Response

Comments and Suggestions for Authors

General comments: The paper covers an interesting and relevant topic; the environmental risk assessment of wetlands. The procedure used seems adequate, with a large number of samples collected and analyzed, and the use of indicators to perform the risk assessment. However, the quality of English language makes it difficult to fully comprehend the article. A thorough revision by a native English speaker is required. Noted and done accordingly. The discussion section should also be enhanced, see specific comments for questions. Some conclusions require scientific evidence to prove they are more than speculations. For these reasons, I recommend to accept the paper with major revision.                                      Duly noted and such discussions are added/mentioned in 15 new lines: 326-333, 446, 558-560, 565-566, 574, 577-581 as well as in the conclusion section.

Lines 46-54: Include more references on wetlands and their use to monitor pollution.                                                                                                   Done, 5 more references added, lines 56-58

Line 74 (and elsewhere): Do not refer to the position of the figure relative to the text (below), as the figure may be placed elsewhere in the final version.                                                                                                          Done, new line number/s:  78, 90, 317

Lines 78-79: Does the National Wetland Park status preclude other land uses? How can such a protected area be as damaged?                                                                                                                                                      This area was listed as National wetland park in 2012. In the past 50 years, most parts of it were changed to farmland and fish ponds by locals, new line number/s:  79-81

Line 87: Why is the sample depth different for soil and sediments?                    The sample depth in both soil and sediments was 0-10cm and it is amended in the content too, new line number/s:  91 and 93

Line 90: What are external impacts?                                                                  It can protect them from rainfall effects, keep them separated, proper labeling process,.. etc. it is addressed in the manuscript, new line number/s:  109-110

Lines 101-102: Specify which type of digestion was used for which elemental analysis.                                                                                                          It is clarified and specified (About 0.5g dry sediment/ soil for each sample was digested in a microwave oven with a mixture of acid H2O2-HNO3-HF-HCl (120℃/3min-150℃/5min-170℃/25min) for Cu, Cr, Cd, Ni, Zn and Pb, and about 0.1g dried of each sample was digested using mixed acid of HNO3 and HCl at 100℃ for 2h then diluted to 25ml with distilled water for As and Hg.), new line number/s:  115-119

Line 105: What do you mean by “control the ambient conditions”?                   Added (such as temperature and humidity in the lab during the testing process), new line number/s:  127

Line 106: The reported relative standard deviation corresponds to which analysis, or which contaminant?                                                                                  The relative standard deviation (RSD) corresponds to the concentration of standard materials for all the contaminants (GBW07404), which is calculated by:(the formula can be found in the attached response file), new line number/s: 128-130

Line 133: Specify the limits for each category.                                                                                                         Done, new line number/s:  277-281

Line 139: Is the “pollutant concentration standard” the background concentration?                                             It refers to the standard value of that particular contaminant, new line number/s:  285

Line 141: Specify the limits for the three other categories.                                                                                   Done, new line number/s:  287-289

Lines 153-155: Use the same symbols as in the other equations, it would be easier for the reader to follow.                                                                               Done accordingly in 1st and 4th equations, new line number/s:  274 and 299

Line 162: The ANOVA results are not shown.                                                                                                       ANOVA analysis is shown in table 1, it shows how significantly the concentration of each pollutant is varied in different landuse types.

Line 166: The median concentrations are not shown.                                                                                                   It is mean and corrected, new line number/s:  325

Line 177: “highest element in the Huixian wetland”: no, other elements are much higher!                                                                                                   Although few elements were high in some landuse types, this was to emphasize the importance of Cd as no other element was higher than standard values for all the landuse types, at the same time, new line numbers:  461-462

Line 180: “magnificient”: wrong word.                                                                                                               Changed, new line number/s:  456

Lines 192 and 195: Nickel, not nickol                                                                                                                  Corrected, new line number/s:  476

Lines 195-196: “fewer concentrations”, wrong words                                                                                   Corrected, new line number/s:  479

Line 205: Too many capital letters                                                                                                                               Changed, new line number/s:  500-502

Figure 5: Add category thresholds on graph (similar to Figure 4).                                                                        Done, new line number/s:  figure 5

Lines 254-261: Discuss the relationship (or lack of) between risk and SEPA limits. Do the limits help protect sufficiently against environmental risks?                                                                                                                                                      Added (The environmental standards (SEPA) which were made to promote ecology and improve people's health, besides social and economic developments, can limit the health risks to some extent. Scenarios like Huixian wetland shows that parallel with strengthening the standards, monitoring mechanisms are equally important to guarantee the application of the permitted limits.), new line number/s:  577-581

Lines 262-268: How does the Park status help in management and remediation prioritization?                                                                                               The park management could work with the local government to arrange seasonal remediation to minimize the damages after main contaminant identification and conduction extensive researches, new line number/s:  573-574

Line 272: “identify their possible sources”: the sources were not discussed much, only speculations.                                                                                           PCA analysis helped to introduce the possible sources as much as the research data allows and provides. Therefore we could not go much deeper into the details especially in conclusion. Both fertilizer and limestone bedrocks added, new line number/s:   600-601

Line 276: “due to plant uptake”: not demonstrated.                                                                                                This was a possible reason behind the averagely lower concentration of heavy metals in soil (than sediment) as they are taken mainly from the planted landuse, We did briefly explained this in 329, 553-556,  559 lines.

Line 277: “anthropogenic activities”: speculated.                                                                                                 Noted and mentioned as a possible origin (fertilizer), new line number/s:  601

Reviewer 3 Report

Suggested notes for author:

Check references and format according the instructions for authors.

In addition, part of Materials and methods better specify the QA/QC procedures, add analytical limits of detection and quantitation (LOD, LOQ) and the operating parameters of analytical instrumentation.

Add other statistical descriptive characteristics, concentration is nominal or real?

Can you add background concentration of metals.

Author Response

Suggested notes for author:

Check references and format according the instructions for authors.                      Duly noted and done accordingly.

In addition, part of Materials and methods better specify the QA/QC procedures, add analytical limits of detection and quantitation (LOD, LOQ) and the operating parameters of analytical instrumentation.                                                              The analytical equipment information and detection limits are added, new line numbers: 120- 124

Add other statistical descriptive characteristics, concentration is nominal or real?                                            The concentrations for different metals are real ones, new line number/s:  122, 274

Can you add background concentration of metals.                                                                                                        Noted and It is added (mentioned) in the last row of table 1.

Round 2

Reviewer 2 Report

I would like to thank the authors for addressing my comments and concerns about the manuscript. The revised version is improved. There are still some awkward sentences and formulations, a further English review is required. Two specific (minor) comments:

Line 290: What is BMPs?

Conclusions: this section should be numbered 4 (not 5).

Reviewer 3 Report

Authors corrected manuscript according the comments.